# Solar-Driven Thermocatalytic Synthesis of Octahydroquinazolinone Using Novel Polyvinylchloride (PVC)-Supported Aluminum Oxide (Al_2_O_3_) Catalysts

**DOI:** 10.3390/ma16072835

**Published:** 2023-04-02

**Authors:** Abdulrahman I. Alharthi, Mshari A. Alotaibi, Amani M. Alansi, Talal F. Qahtan, Imtiaz Ali, Matar N. Al-Shalwi, Md. Afroz Bakht

**Affiliations:** 1Chemistry Department, College of Science and Humanities Studies, Prince Sattam Bin Abdulaziz University, P.O. Box 83, Al-Kharj 11942, Saudi Arabia; 2Chemistry Department, King Saud University, Riyadh 12372, Saudi Arabia; 3Physics Department, College of Science and Humanity Studies, Prince Sattam Bin Abdulaziz University, P.O. Box 83, Al-Kharj 11942, Saudi Arabia; 4Preparatory College, Prince Sattam Bin Abdulaziz University, P.O. Box 83, Al-Kharj 11942, Saudi Arabia; 5Electrochemical Sciences Research Chair (ESRC), Department of Chemistry, College of Science, King Saud University, P.O. Box 2455, Riyadh 11451, Saudi Arabia

**Keywords:** polyvinylchloride, aluminum oxide, catalyst, octahydroquinazolinone, solar-powered thermocatalytic reactor

## Abstract

The chemical industry is one of the main fossil fuel consumers, so its reliance on sustainable and renewable resources such as wind and solar energy should be increased to protect the environment. Accordingly, solar-driven thermocatalytic synthesis of octahydroquinazolinone using polyvinylchloride (PVC)-supported aluminum oxide (Al_2_O_3_) as a catalyst under natural sunlight is proposed in this work. The Al_2_O_3_/PVC catalysts were characterized by FT-IR, SEM, BET, XRD, and XPS techniques. The obtained results indicate that the yield and reaction time can be modified by adjusting the molar ratio of the catalyst. To investigate the stability of the catalyst, the spent catalyst was reused in several reactions. The results indicated that, when a 50% Al_2_O_3_ catalyst is employed in an absolute solar heat, it performs exceptionally well in terms of yield (98%) and reaction time (35 min). Furthermore, the reaction times and yield of octahydroquinazolinone derivatives with an aryl moiety were superior to those of heteroaryl. All the synthesized compounds were well characterized by FT-IR, ^1^H-NMR, and ^13^C-NMR. The current work introduces a new strategy to use solar heat for energy-efficient chemical reactions using a cost-effective, recyclable environmentally friendly PVC/Al_2_O_3_ catalyst that produces a high yield.

## 1. Introduction

As the chemical industry requires considerable amounts of energy, its reliance on fossil fuels needs to be reduced to protect the environment. As solar energy is readily available and does not generate any harmful byproducts, it is a viable alternative [1]. In extant studies, chemical reactions induced by solar radiation have already been explored [2,3], indicating that different reagents can be successfully activated by specific wavelengths in the solar spectrum. However, the mechanisms currently employed for this purpose are complex, expensive, difficult to monitor, and insufficiently selective [4]. Moreover, in most cases, useful molecules can only be synthesized if the incident radiation can be tuned to be efficiently absorbed by the materials being used. Since many substances do not absorb radiation in the visible part of the solar spectrum, or require specific wavelengths from the UV region, the use of natural light for chemical processes is often impractical. Nonetheless, as sunlight generates considerable heat, it can promote dependable and sustainable chemical synthesis [5] without generating any harmful byproducts [6]. These benefits are already recognized, as organic transformations induced by insolation have already been used in the synthesis of new chemical bonds [7]. Heterogeneous catalysts are of particular interest in this context, as these chemical compounds are physically distinct from the reactants and/or products involved in the catalyzed chemical reaction [8].

In extant research, solid-phase heterogeneous catalysts, such as aluminum oxide, are widely employed to expedite the chemical reaction between two reactants [9,10] due to their beneficial chemical properties and mechanical stability [11]. Given that the acidic properties of aluminum oxide as a single (Al_2_O_3_), binary (SiO_2_-Al_2_O_3_, Al_2_O_3_-TiO_2_), and tertiary (TiO_2_-SiO_2_-Al_2_O_3_) oxide in different organic reactions are well understood [12], its different forms are economically attractive catalysts for a wide variety of organic reactions [13], as they are capable of inducing a wide range of acid-catalyzed reactions [12]. However, metal leaching that typically accompanies this process prevents the reuse of these oxides, making them too expensive for large-scale applications. These issues can be overcome by employing inorganic (calcium carbonate, calcium kaolinites) and organic (such as polystyrene or divinylbenzene) polymers for promoting organic reactions [14,15,16,17,18]. The most optimal solid organic polymers for this purpose are chosen based on their stability and recyclability, as well as whether they bind with the substrate/reagent covalently or non-covalently [16]. Accordingly, polyvinylchloride (PVC) is frequently used for solid-phase synthesis as it is inexpensive and versatile [19], even though its thermal stability and processibility are inferior to that of polyethylene, polypropylene, and polyamide [20]. These shortcomings could be overcome by combining PVC with organic and inorganic fillers such as calcium carbonate and calcium kaolinites [20,21], as well as by chelating it with metal ions such as Co^+2^, Ti^+3^, Cr^+3^, Fe^+3^, Cu^+2^, and Zn^+2^ [22,23]. Thus, Al_2_O_3_ could be supported by a PVC matrix to improve its binding with a polymeric partner and enhance the thermocatalytic synthesis selectivity and yield.

Numerous medicinally effective substances such as antimicrobial [24], anti-inflammatory [25], anticancer [26] and antiviral [27] drugs contain octahydroquinazoline and its derivatives, as these are cost-effective substances characterized by high selectivity. In recent years, several multicomponent techniques for the synthesis of octahydroquinazolinone derivatives have been proposed [28], including Biginelli reactions [29] which rely on the condensation of carbonyl compounds and urea in the presence of multiple Lewis acid catalysts to synthesize quinazolines [30,31,32,33,34]. The greatest advantage of this process is that octahydroquinazolinone can be synthesized through simultaneous condensation of dimedone, aldehyde, and urea/thiourea with Lewis acid, protic acid, and solid acid under traditional reflux and solvent-free conditions by exposing the components to microwave and ultrasonic irradiation [35]. However, the high acidic potential of these catalysts makes them difficult to handle, pointing to the need for more environmentally friendly alternatives [36].

Thus, research in this field is increasingly shifting toward solar heat as its use would reduce the number of reaction steps [37] while resulting in more environmentally friendly chemical processes [6]. To date, several photo-assisted mechanistic approaches for the synthesis of quinazoline/octahydroquinazolinone have been proposed, such as photocatalysis, energy transfer, proton-coupled electron-transfer system, and hydrogen atom transfer [38]. We have envisaged with our previous efforts using recyclable catalysts and conventional heating methods for the synthesis of octahydroquinazolinone/quinazoline synthesis [39,40]. To contribute to this ongoing endeavor, in this work, we report on a novel three-component reaction for solar irradiation-induced octahydroquinazolinone synthesis based on aluminum oxide supported by PVC as a thermocatalyst, as shown in Figure 1.

## 2. Materials and Methods

### 2.1. Materials

Aluminum oxide was purchased from Loba Chemie (Mumbai, India), while a fine PVC (Polyvinyl Chloride) powder with a 36 μm particle diameter was supplied from SABIC corporation (Riyadh, Saudi Arabia) as a support material for the catalyst. Dimedone (5,5-Dimethyl-1,3-cyclohexanedione) (CAS No.126-81-8), Urea (CAS No. 57-13-6), Thiourea (CAS No. 62-56-6), and aldehydes such as 4-Chlorobenzaldehyde (CAS No. 104-88-1), 4-Fluorobenzaldehyde (CAS No. 459-57-4), 4-Hydroxy-3-methoxybenzaldehyde (Vanillin) (CAS No. 21-33-5), Furfural (Furan-2-carboxyaldehydes) (CAS No. 98-01-1) were acquired from Sigma Aldrich (St. Louis, MO, USA).

### 2.2. Catalyst Characterization

Thermo Science’s iD5 ATR diamond Nicolet is 5 FT-IR Spectrometer was used to record the Fourier transform infrared (FT-IR) spectra, while Cu Kα radiation (λ = 1.543 Å) provided by an X-ray diffractometer (Ultima IV, Rigaku, Japan) in the 10°−80° 2θ range was used to capture X-ray diffraction (XRD) spectra to define the phase composition of produced catalysts. The morphological properties of all studied samples were assessed using a field emission scanning electron microscope (FESEM, Model: Quanta FEG 250, Thermo Fisher Scientific, Amsterdam, The Netherland), and their chemical composition was characterized using Thermo scientific K-alpha X-ray photoelectron spectrometer (XPS) Waltham, MA, USA with a characteristic energy of 1486.6 eV generated by a monochromic Al Kα source. Throughout the XPS measurements, a pressure of approximately 10^−8^ m bar, room temperature (RT), and 400 µm spot size were maintained. XPS survey scans used for elemental identification were obtained at 200 eV pass energy and 1 eV step size, while 50 eV and 0.1 eV were employed for capturing high-resolution XPS images. The catalyst’s BET surface area, pore radius, and pore volume were estimated by N_2_-physisorption at 77 K using Quantachrome ASiQwin software, version 5.2. Finally, Bruker-Plus (400 MHz) nuclear magnetic resonance (NMR) apparatus was used to record the ^1^H-NMR and ^13^C spectra of synthetic octahydroquinazolinones with tetramethylsilane serving as an internal reference.

### 2.3. Al_2_O_3_/PVC Catalyst Preparation

Catalysts were prepared by the wet impregnation method. For the preparation of a 5% Al_2_O_3_ catalyst, 50 mL of distilled water was mixed with Al_2_O_3_ and PVC (at 5:95 wt. ratio) in a 100 mL beaker and the contents were rapidly agitated at 100 °C for up to one hour. The resulting catalyst was then stored inside a hot air oven overnight at 90 °C. The same process was adopted to obtain Al_2_O_3_/PVC catalysts containing 25%, 50%, 60%, and 75% Al_2_O_3_ (henceforth denoted as 25% Al_2_O_3_, 50% Al_2_O_3_, 60% Al_2_O_3_, and 75% Al_2_O_3_) all of which subjected to XRD, FT-IR, Brunauer–Emmett–Teller (BET), and scanning electron microscopy (SEM) analyses, along with pure PVC and Al_2_O_3_ samples to facilitate comparisons.

### 2.4. Generalized Method for the Synthesis of Octahydrquinazolinone Derivatives

For the synthesis of octahydroquinazolinone derivatives, dimedone, urea/thiourea, and various aldehydes were mixed with an optimized amount of ethanol and 100 mg of Al_2_O_3_/PVC catalyst in a 100 mL beaker placed on a magnetic stirrer. Heat (75–80 °C) generated by sunlight was regularly measured a by thermometer and slight solvent evaporation was compensated for during the reaction. Before its use, the beaker was painted black to facilitate the absorption of the heat generated by sunlight. Once the beaker was painted can be reused many times in several reactions. Mixtures in a ratio of 7:3 of acetone and ethyl acetate were used as the solvent, and the reaction progress was regularly monitored. As the aim was to reuse the solid catalyst in subsequent reactions, the organic layer was separated by centrifugation, after which the solid product was purified by evaporation and recrystallization with ethanol. The final compounds were characterized by determining the melting point, as well as by analyzing the FT-IR, ^1^H NMR, and ^13^C NMR spectra.

### 2.5. Spectroscopic Data of the Newly Synthesized Compound

4-(furan-2-yl)-7,7-dimethyl-2-thioxo-1,2,3,4,5,6,7,8-octahydroquinazoline-2-thione. White solid; FT-IR (cm^−1^, ATR); 3429 and 2952 (NH), 1583 (C=O, ring), 1456 (C=S, thiourea), 1372 (C=C); ^1^H NMR (DMSO-d6, 400 MHz): 10.66 (s, 1H, NH), 7.39 (s, 1H, NH), 6.27–6.25 (s, 1H, *J* = 8.16 Hz, Ar–H), 6.27–6.25 (s, 1H, *J* = 8.16 Hz, Ar–H), 5.85–5.37(2H, m, Ar–H), 5.19 (1H, s, CH), 2.50 (d, 2H, *J* = 1.44 Hz, CH_2_), 2.28(s, 2H, CH_2_), 1.01 (s, 6H, 2 × CH_3_); ^13^C NMR (DMSO-d6, 100 MHz): δ 187.50 (C=O), 141.46 (NC=C), 113.42, 110.43, (ArC), 105.43 (OC−C=C), 46.97(1C, CH_2_), 31.86 (1C, CH_2_), 28.03 (1C, CH), 26.89 (2CH_3_). 

## 3. Results and Discussion

### 3.1. Characterization

#### 3.1.1. X-ray Diffraction (XRD) Measurements

Figure 2 shows the XRD patterns of raw PVC, Al_2_O_3,_ and prepared Al_2_O_3_/PVC catalysts. The XRD pattern of PVC powder displays relatively broad peaks, indicating its amorphous character [41]. However, crystallinity starts to emerge as the Al_2_O_3_ content in the Al_2_O_3_/PVC catalyst increases from 25% to 75%. The XRD patterns of the prepared samples, Al_2_O_3_/PVC catalysts, demonstrate that a mixture comprising 67.3% aluminum oxide, 16.7% bassanite (2CaSO_4_·H_2_O), 2.5% calcium sulfate, 10.3% aluminum oxide hydroxide (boehmite), and 3.2% aluminum hydroxide oxide hydrate (nordstrandite) as the raw material reacts with water molecules from the air or the aqueous medium used in sample preparation.

#### 3.1.2. Fourier Transform Infrared (FT-IR) Measurements

The FT-IR spectra of aluminum oxide, polyvinyl chloride, and Al_2_O_3_/PVC catalyst with different aluminum oxide ratios are displayed in Figure 3. As can be seen from the graph, the main infrared peaks produced by the aluminum powder are located at ~594 and ~465 cm^−1^, which corresponds to the Al−O bending vibration of Al–OH groups. In addition, the peak at 652 cm^−1^ represents Al−O stretching vibration.

Al_2_O_3_ FT-IR spectra show bands of absorption at 3606 cm^−1^ that are corresponding to the stretching vibrations of the O–H groups and water [42]. On the other hand, PVC produces peaks at 2913 cm^−1^, ~1425 cm^−1^, 1324 cm^−1^, 1088 cm^−1^, 957 cm^−1^ and 614 cm^−1^, respectively, reflecting the –CH_2_- asymmetric stretching vibration, wagging –CH_2_, CH_2_ deformation, C−H stretching from CH−Cl, rocking CH_2_, and C−Cl stretching [43]. The obtained spectra further reveal that, as the Al_2_O_3_ content in the sample increases from 5% to 75%, several PVC peaks diminish and can no longer be discerned might be due to Al_2_O_3_ concentration overlapping the PVC peaks.

#### 3.1.3. Structural Properties

Table 1 displays the surface area analysis of the pure PVC, Al_2_O_3_, and Al_2_O_3_/PVC catalysts with different amounts of Al_2_O_3_ in the range of 5–75 wt.%. It can be noted that pure PVC has a very low surface area of 3.70 m^2^/g, whereas pure Al_2_O_3_ has a high surface area of 105.40 m^2^/g. In addition, it can be seen that increasing the amount of Al_2_O_3_ in Al_2_O_3_/PVC catalyst enhances the surface area of the catalyst and pore volume where they increased from 7.30 up to 76.50 m^2^/g and from 0.017 up to 0.182 cc/g, respectively. Additionally, the pore radius was observed to increase with increasing the amount of Al_2_O_3_ to 50 wt.% and then decreased with 60 and 75 wt.% of Al_2_O_3_. The maximum pore radius was noted using a 50 wt.% Al_2_O_3_/PVC catalyst where it was 32.70 Å. However, it seems that there is an integration between Al_2_O_3_ and PVC that has led to an increase in the surface area, as Al_2_O_3_ has a higher surface area than PVC.

Figure 4a,b presents the N_2_ adsorption-desorption isotherms and pore size distributions of the catalysts. According to the IUPAC standard, all catalysts exhibited a type V isotherm with an H3 hysteresis loop as shown in Figure 4a. This type of hysteresis revealed that the catalysts have mesoporous structures. However, type H3 hysteresis is generally found on solid materials that consist of aggregates or agglomerates of particles forming slit-shaped pores (plates or edged particles such as cubes), with irregular size and/or shape [44]. Moreover, it can be seen that the increase in the amount of Al_2_O_3_ added to the PVC led to higher N_2_ adsorbed volume in the p/p° range of 0.49 to 0.95. This finding is consistent with the results shown in Table 1, where the surface area and pore volume increased with increasing the amount of Al_2_O_3_. Figure 4b illustrates the pore size distributions. It was reported that solid materials which have pore sizes in the range from 2 to 50 nm classify as mesoporous [44]. From Figure 4b, it can be noted that increasing the Al_2_O_3_ content produced pore sizes well within the mesoporous range, where the pore size distribution was with a peak at 40 nm. The pure PVC sample showed a very low number of pores, and they increased as a result of the addition of Al_2_O_3_. Therefore, it seems that the Al_2_O_3_ content in the samples plays a significant role in changing the textural properties of the Al_2_O_3_ PVC catalysts.

#### 3.1.4. Scanning Electron Microscopy (SEM) Analysis

The morphology of Al_2_O_3_, PVC, and Al_2_O_3_/PVC samples with different molar ratios of Al_2_O_3_ was analyzed using SEM at different magnifications and the obtained images are shown in Figure 5. As can be seen from Figure 5a,g, PVC comprises sporadic mushroom cap spores, while the Al_2_O_3_ structure is characterized by the agglomeration of fine particles of various shapes (rectangles, oval bars, and cubes) and sizes. It is also apparent that the Al_2_O_3_ particle size is smaller than that of pure PVC. Consequently, the Al_2_O_3_/PVC catalysts with 5 to 75% Al_2_O_3_ content exhibit a gradual shift in morphology, becoming progressively more similar to Al_2_O_3_, as shown in Figure 5c−f. As surface area is inversely proportional to the particle size [45], increasing the amount of Al_2_O_3_ in the Al_2_O_3_/PVC catalyst increases the surface available for chemical reaction. This is well-aligned with the BET surface area results, in Table 1.

#### 3.1.5. X-ray Photoelectron Spectrometry (XPS) Analysis

XPS survey spectra were captured to study the surface chemistry and the chemical composition of PVC, Al_2_O_3,_ and Al_2_O_3_/PVC catalysts with different Al_2_O_3_ amounts, as indicated in Figure 6a. It is evident that both PVC and Al_2_O_3_/PVC contain C, Cl, and O. Additionally, the Al_2_O_3_/PVC catalysts contain Al, Ca, and S. However, the Al_2_O_3_ sample comprises C, O, Al, Ca, and S. The presence of oxygen in the XPS survey spectra of PVC might be ascribed to contamination or polymer chain oxidation [46]. Similarly, the presence of both Ca and S in the Al_2_O_3_/PVC spectra is expected, because the Al_2_O_3_ sample contains CaSO_4_·H_2_O. The high-resolution C 1*s* spectra produced by PVC, Al_2_O_3_/PVC catalysts, and Al_2_O_3_ are shown in Figure 6b, respectively. The PVC sample comprises three kinds of carbon, which produce peaks at 287.13 eV, 288.52 eV, and 289.54 eV binding energies that correspond to the C–C and C–H and the C–Cl bonds and O=C−O bonds, respectively [46,47]. Additionally, catalysts containing smaller amounts of Al_2_O_3_ exhibit three peaks at around 285.5, 287.5, and 289.6 eV, reflecting the presence of C−C, C−Cl, and O=C−O bonds. However, as the Al_2_O_3_ content in the Al_2_O_3_/PVC catalyst increases, C–C and C–Cl peaks decrease in amplitude, while the magnitude of O–C=O peaks increases. The C 1*s* XPS spectra of the 75% Al_2_O_3_ catalyst and Al_2_O_3_ samples can be deconvoluted into two peak components at around 286.6 and 289.7 eV binding energies, which are attributed to the C−C and O=C−O species, respectively. 

Figure 6c shows high-resolution C1 2*p* spectra produced by PVC, Al_2_O_3_/PVC catalysts, and Al_2_O_3_. The PVC spectrum can be decomposed into three peaks, namely Cl 2*p*_1/2_ at 204.7 eV and peak Cl 2*p*_3/2_ at 202.6 eV representing organic chlorine atoms covalently bounded sp2 carbon [46,47], whereby the latter is assigned to the chloride ion and the hydrogen bonds [48]. On the other hand, Cl 2*p* spectra produced by most Al_2_O_3_/PVC catalysts contain two peaks of high intensity with a maximum of 203.2 eV (Cl 2*p*_1/2_) and 200.1 eV (Cl 2*p*_3/2_), which are attributed to the C–Cl bonds. The oxygen O 1*s* spectra of the PVC, Al_2_O_3_/PVC catalysts, and Al_2_O_3_ sample are also presented in Appendix A where the two peaks at 533.1 and 535.2 eV, which are attributed to the C–C=O and C–O–H bonds, respectively, characterize the XPS O 1*s* spectrum produced by PVC [46]. On the other hand, the O 1*s* peak produced by most of the Al_2_O_3_/PVC catalyst samples can be decomposed into three peaks located at around 532 and 534 eV, corresponding to O bound to the Al lattice (Al−O−Al bonds) and OH/COO bonds, respectively [49]. The evidence of OH/COO bonds in the Al_2_O_3_ spectra is attributed to the use of H_2_O as the reaction medium in the present study and is consistent with the findings reported by other authors [50,51]. Similarly, the peak at 536 eV arises due to the adsorption of free water molecules. As noted above, the intensity and number of O 1*s* peaks increase, while the intensity and number of C 1*s* and Cl 2*p* peaks decrease with the increase in Al_2_O_3_ content in the Al_2_O_3_/PVC catalysts, confirming that adding Al_2_O_3_ introduces abundant oxygen atoms into the PVC chain.

Appendix A shows the Al 2*p* spectra produced by Al_2_O_3_, Al_2_O_3_/PVC catalysts, and PVC. The Al_2_O_3_/PVC samples contain the two peaks produced by Al 2*p* of Al_2_O_3_ (at around 77 and 79 eV) confirming the presence of Al–O and Al–OH bonds, respectively [52]. Additionally, the peak located at around 75 eV is attributed to the AlO(OH) bond. In the spectrum produced by the 50% Al_2_O_3_ catalyst sample, a greater contribution from the OH groups is evident [50,51]. Thus, as demonstrated by the XPS results, which are in good agreement with the XRD and FT-IR data, Al_2_O_3_ was successfully mixed and dispersed inside the PVC matrix to create Al_2_O_3_/PVC catalysts. Furthermore, Appendix A with the ratios of the electronic state of the elements for the as-prepared Al_2_O_3_/PVC catalysts was displayed in the Appendix A.

### 3.2. Catalytic Activity

To achieve optimized reaction conditions, the catalyst and solvent amounts, as well as the reaction, were modified, and the findings are reported in Table 2, Table 3, Table 4, Table 5 and Table 6.

When ethanol was used without a catalyst, no reaction was observed after 12 h. Moreover, experiments conducted with different Al_2_O_3_ amounts revealed that the reaction involving 50% Al_2_O_3_ resulted in the highest (98%) yield while requiring only 35 min to complete, which could be attributed to the sufficient number of Brönsted and Lewis acid sites [3], as reflected in a greater pore radius, Table 1. Thus, as this Al_2_O_3_ quantity in the PVC matrix may be led to the optimal development of Lewis acid-base interactions between the polar surface group of the Al_2_O_3_ and the ionic species of the PVC, it can be adopted to improve the ionic conductivity, thermal conductivity, and mechanical stability of the 50% Al_2_O_3_ catalyst. Consequently, in the subsequent experiments, 50% Al_2_O_3_ was used as a catalyst, but its quantity was varied in the 20–120 mg range, to assess the influence of these factors on the reaction efficiency. As can be seen from Table 3, 100 mg of catalyst yields the most optimal results.

Next, 100 mg of 50% Al_2_O_3_ catalyst was used while varying the solvent type to assess its influence on the reaction efficiency. As can be seen from Table 4, ethanol is most conducive for high yields, as the values obtained for CH_2_Cl_2_, DMF, MeOH, EtOH, H_2_O, and EtOH/H_2_O are much lower and the reactions took longer time to complete, and also the amount of solvent in mL was also wasted in each type during sunlight irradiation depending upon the boiling point of solvents. Therefore, each solvent was continuously added as required during the course of the reaction.

To ensure that 50% Al_2_O_3_ is the most beneficial catalyst, additional experiments were conducted using HCl (Conc.), H_2_SO_4_ (Conc.), *P*-TsOH, Al_2_O_3,_ and PVC and the results are presented in Table 5.

The reactions based on the 50% Al_2_O_3_ catalyst require the shortest time to complete while producing the highest yield, confirming that Al_2_O_3_/PVC has superior catalytic potential compared to all other tested compounds, including aluminum oxide and polyvinyl chloride (PVC).

Finally, the optimal conditions established through previous experiments (100 mg of 50% Al_2_O_3_ as a catalyst, optimized amount of ethanol) were adopted for the synthesis of 4-(substituted)-7,7-dimethyl-1,2,3,4,5,6,7,8-octahydroquinazoline-2,5-dione to assess the adaptability of the proposed protocol to other processes. Various substituents on aldehyde including Cl, F, OCH_3_, OH, and Furan moieties were used for this purpose and the obtained results are reported in Table 6.

As evident from the tabulated results, 50% Al_2_O_3_ outperformed all considered compounds in terms of both generated yield and reaction time. 

### 3.3. Plausible Octahydroquinazolinone Production Mechanism Using Al_2_O_3_/PVC as a Catalyst

As shown in Figure 7, Al_2_O_3_/PVC activates the carbonyl group of the aldehyde to produce Intermediate I under the influence of solar heat. This is followed by condensation with urea/thiourea (Intermediate II) and subsequent dehydration, resulting in Intermediate III, which reacts with dimedone to produce Intermediate IV. Finally, cyclization with the removal of water yields the desired octahydroquinazolinones.

### 3.4. Recyclability of the Al_2_O_3_/PVC

As sufficient catalyst reusability and recovery are crucial for obtaining sustainable and environmentally friendly synthesis methods that can be adopted in practice, these aspects were also considered in the present study, and the findings are reported in Table 7. The catalyst can be reused up to four times without a significant loss in yield and most of its original mass can be recovered by centrifugation. 

## 4. Conclusions

In this work, functionalized 1,2,3,4,5,6,7,8-octahydroquinazolinone derivatives were successfully synthesized through an affordable process that relies on polyvinylchloride-supported aluminum oxide through sunlight exposure as a free source of heat. As the catalyst can be reused, this environmentally friendly method can be adopted in a wide range of processes, given that it produces a higher yield in a shorter time compared to the previously reported conventional thermal technique. The FT-IR and XRD results indicated the formation of the Al_2_O_3_/PVC catalysts. Additionally, the obtained results indicate that the yield and reaction time can be modified by adjusting the molar ratio of the catalyst. A total of 50% Al_2_O_3_/PVC catalyst performed better than any other alternative under solar heat in terms of yield and reaction time. Additionally, under solar heat, aryl-modified octahydroquinazolinone performed much better than heteroaryl when compared to reaction times and yield. In the context of green chemistry, this study introduces a new strategy towards the use of abundant solar energy for a cost-effective, energy-efficient, and environmentally friendly chemical industry.

## Figures and Tables

**Figure 1 materials-16-02835-f001:**
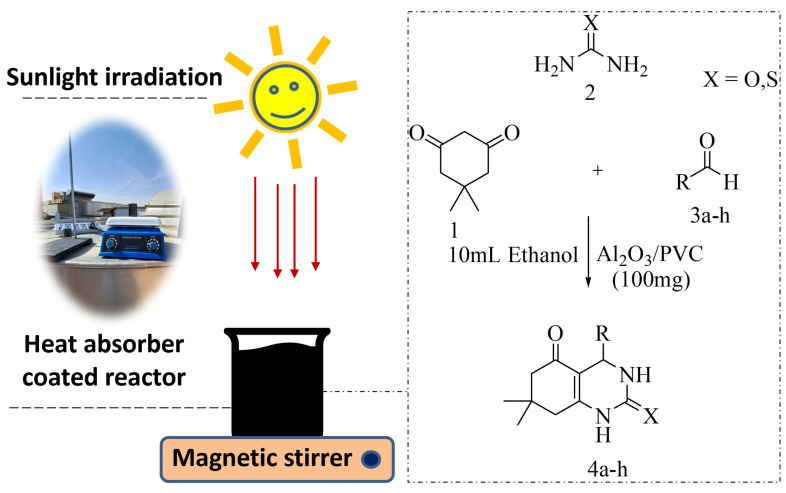
The setup used in the present study for the thermocatalytic synthesis of octahydroquinazolinone by Al_2_O_3_/PVC catalyst.

**Figure 2 materials-16-02835-f002:**
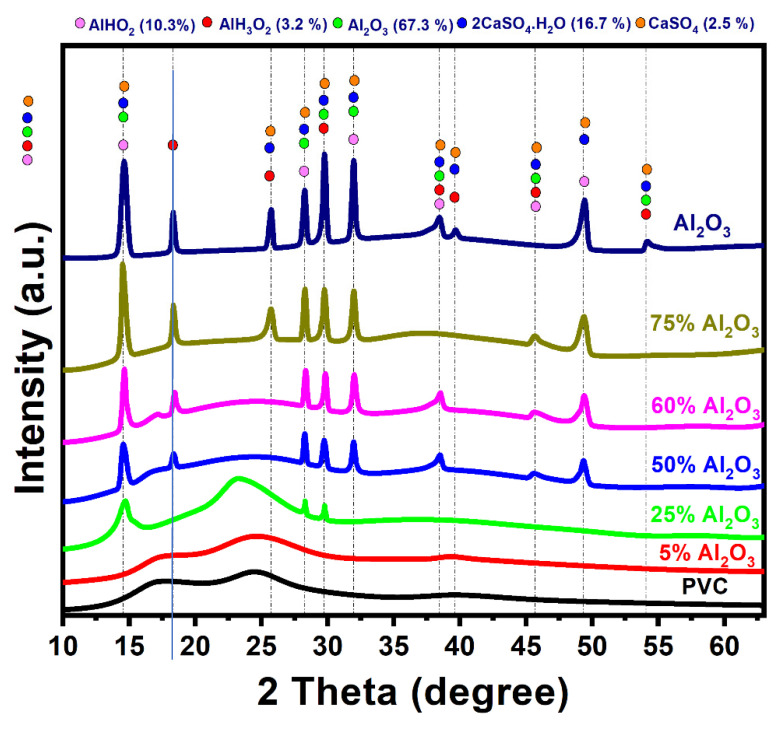
XRD patterns produced by Al_2_O_3_, PVC, and Al_2_O_3_/PVC samples containing different aluminum trioxide amounts.

**Figure 3 materials-16-02835-f003:**
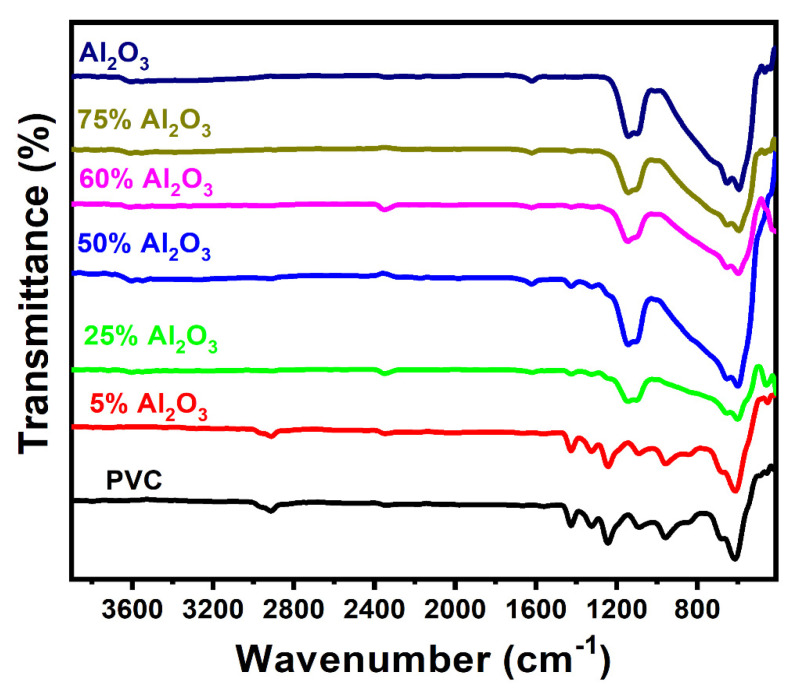
FT-IR spectra of pure Al_2_O_3_, PVC, and Al_2_O_3_/PVC samples containing different aluminum trioxide amounts.

**Figure 4 materials-16-02835-f004:**
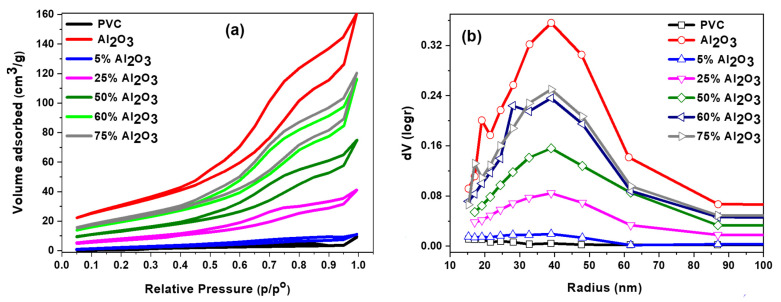
Nitrogen adsorption/desorption isotherms (**a**) and BJH pore size distribution (**b**) of Al_2_O_3_/PVC catalyst with different amounts of Al_2_O_3_.

**Figure 5 materials-16-02835-f005:**
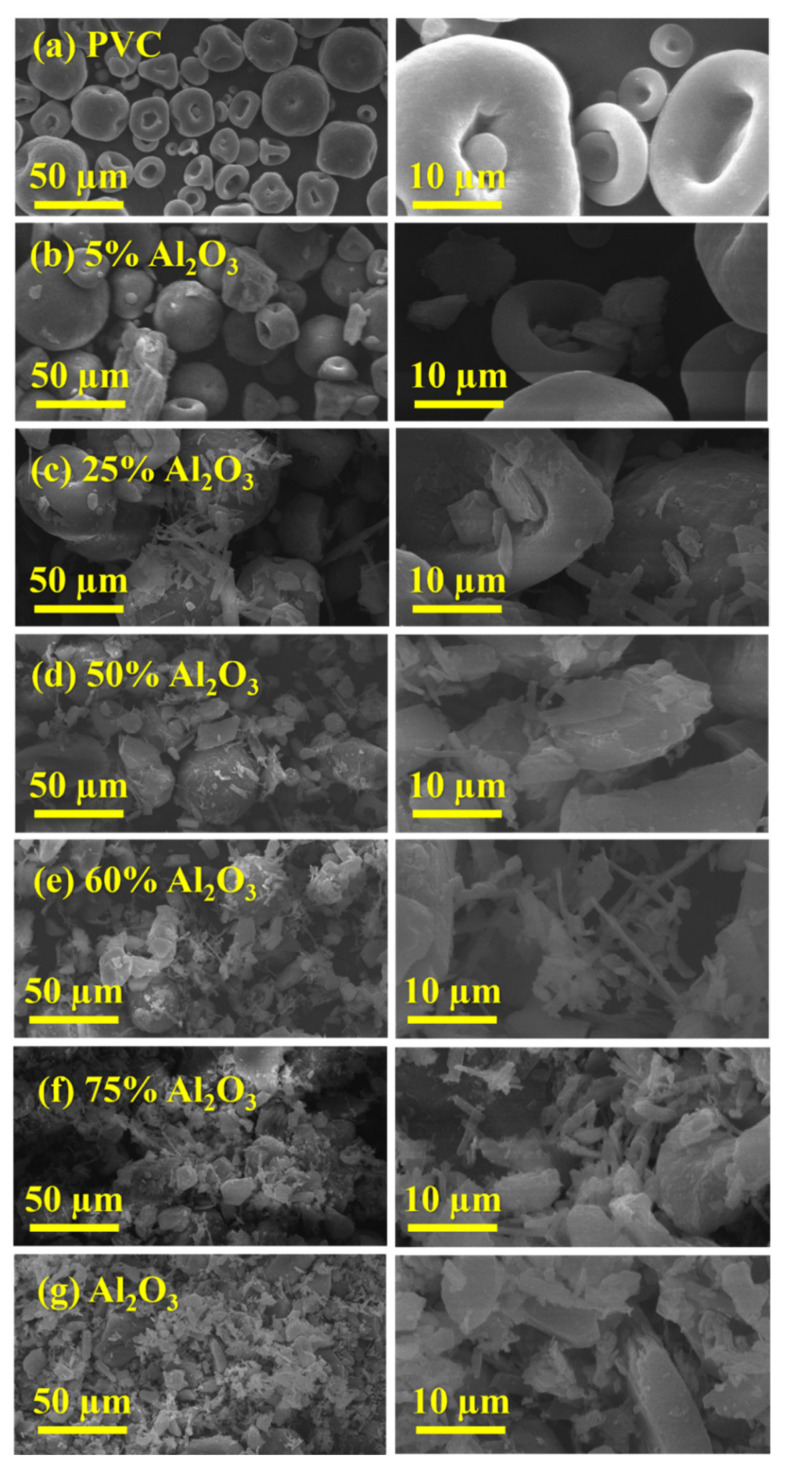
SEM micrograph at different magnifications of (**a**) pure Al_2_O_3_, (**g**) PVC, and (**b**–**f**) Al_2_O_3_/PVC samples containing different aluminum trioxide amounts.

**Figure 6 materials-16-02835-f006:**
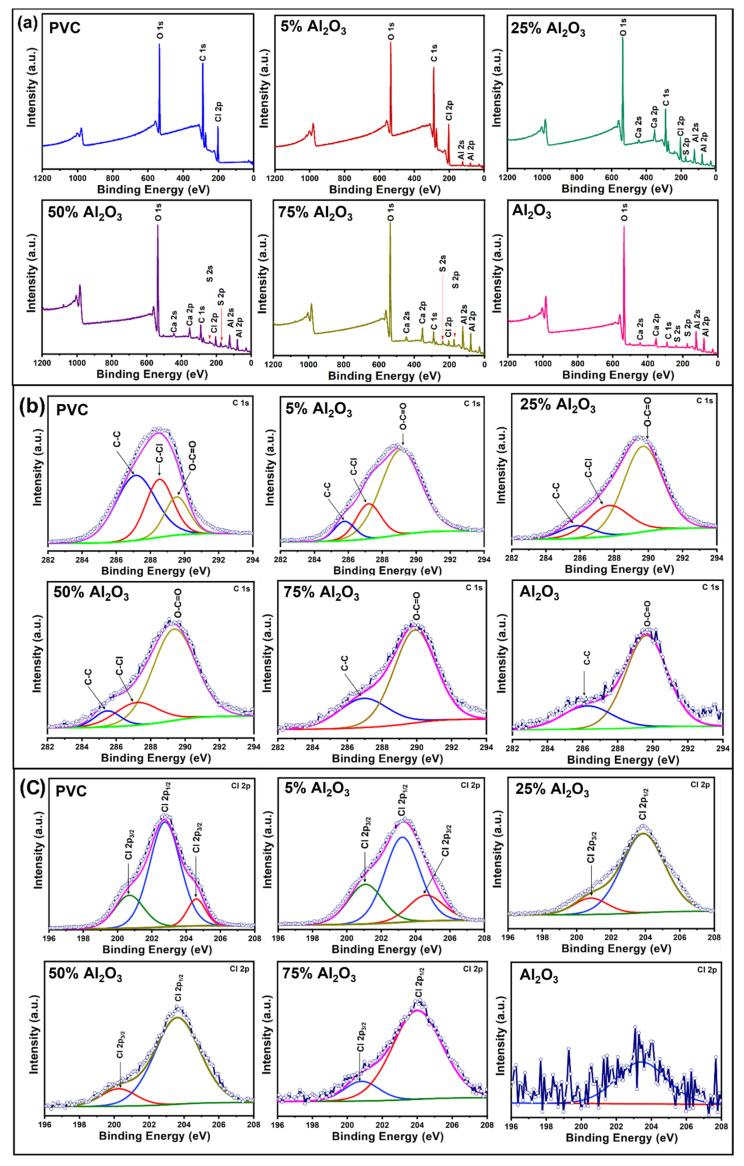
XPS survey (**a**), C 1*s* (**b**), and Cl 2*p* (**c**) spectra of Al_2_O_3_, PVC, and Al_2_O_3_/PVC samples containing different aluminum trioxide amounts as indicated.

**Figure 7 materials-16-02835-f007:**
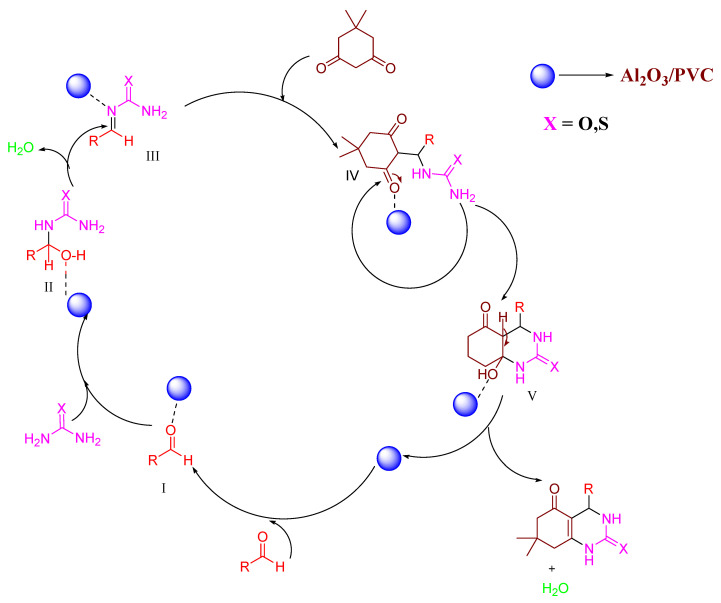
A proposed mechanism for the synthesis of octahydroquinazolinone derivatives using Al_2_O_3_/PVC catalyst.

**Table 1 materials-16-02835-t001:** Textural properties of the Al_2_O_3_/PVC catalysts. Specific surface area pore volume and pore radius.

Catalyst	Surface Area (m^2^/g)	Pore Volume (cc/g)	Pore Radius (Å)
PVC	4	0.014	15.3
Al_2_O_3_	105	0.240	19.1
5% Al_2_O_3_	7	0.017	15.3
25% Al_2_O_3_	25	0.060	28.0
50% Al_2_O_3_	44	0.110	32.7
60% Al_2_O_3_	72	0.176	28.2
75% Al_2_O_3_	77	0.182	17.1

**Table 2 materials-16-02835-t002:** Effects of the Al_2_O_3_/PVC catalyst loaded with different amounts of Al_2_O_3_ on the synthesis of model compound (**4a**) ^a^.

Entry	(%) Al_2_O_3_/PVC	Yield (%) ^b^	Time (h/min)
1	-	Trace	12 h
2	5% Al_2_O_3_	72	2 h
3	25% Al_2_O_3_	82	80 min
4	50% Al_2_O_3_	98	35 min
5	60% Al_2_O_3_	97	45 min
6	75% Al_2_O_3_	94	1 h

^a^ Dimedone (1 mmol), 4-Chloro benzaldehyde (1 mmol), and urea (1.2 mmol) with 100 mg of Al_2_O_3_/PVC as a catalyst in an optimized amount of ethanol. ^b^ Yield of isolated products (bold values are indicative of excellent results under identical reaction conditions).

**Table 3 materials-16-02835-t003:** Effect of 50% Al_2_O_3_ catalyst loading for the synthesis of the model compound under optimized conditions under solar heat.

Entry	Catalyst Amount (mg)	Yield ^a^ (%)
1	20	40
2	40	55
3	60	75
4	80	85
5	100	98
6	120	92

^a^ Isolated yield (values in bold indicate superior outcomes for the same reaction conditions).

**Table 4 materials-16-02835-t004:** Model compound synthesized with different solvents using 100 mg of 50% Al_2_O_3_ catalyst under solar heat ^a^.

Entry	Solvent (mL) ^b^	Yields% ^c^	Time (min)
1	CH_2_Cl_2_	42	300
2	DMF	60	150
3	MeOH	75	60
4	EtOH	98	35
5	H_2_O	55	440
6	EtOH/H_2_O	78	480

^a^ Dimedone (1 mmol), 4-chloro benzaldehyde (1 mmol), and urea (1.2 mmol) with 100 mg of 50% Al_2_O_3_ as a catalyst in an optimized amount of ethanol. ^b^ Amount of solvent was compensated during the experiment for each solvent during the course of the reaction. ^c^ Yield of isolated products (values in bold indicate superior outcomes for the same reaction conditions).

**Table 5 materials-16-02835-t005:** Effects of different catalyst types on the synthesis duration and yield under solar heat.

Entry	Catalyst	Solvents ^a^	Time (h/min)	Yields ^b^ (%)	References
1	HCl (Conc.)	Ethanol	8 h	30	[34]
2	H_2_SO_4_ (Conc.)	Ethanol	10 h	40	[34]
3	*P*-TsOH	Toluene	9 h	Trace	[35]
4	Al_2_O_3_	Ethanol	2 h	40	[36], This work
5	PVC	Ethanol	7.5 h	Trace	This work
6	CaSO4	Ethanol	24 h	Trace	This work
7	50% Al_2_O_3_	Ethanol	35 min	98	This work

^a^ Amount of solvent was compensated during the experiment for each solvent during the course of the reaction. ^b^ Yield of isolated products (values in bold indicate superior outcomes for the same reaction conditions).

**Table 6 materials-16-02835-t006:** The results obtained when using a 50% Al_2_O_3_ catalyst for the synthesis of different octahydroquinazolinone derivatives under solar heat.

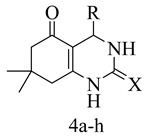									
Entry	R	X		Product	Time (min)	Yield (%) ^a,b^	M.P (°C)		References
							Observed	Reported	
1	4-(Cl)-C_6_H_4_		O	4a	35	98	298–299	301–303	[53]
2	4-(F)-C_6_H_4_		O	4b	40	97	137–138	134–136	[54]
3	4(OH)-3(CH_3_O)-C_6_H_4_		O	4c	50	95	185–187	188–190	[55]
4	Furfural		O	4d	60	93	261–263	262–265	[55]
5	4-(Cl)-C_6_H_4_		S	4e	40	96	222–224	218–220	[56]
6	4-(F)-C_6_H_4_		S	4f	50	95	258–260	260–262	[56]
7	4(OH)-3(CH_3_O)-C_6_H_4_		S	4g	60	92	187–188	188–190	[57]
8	Furfural		S	4h	75	90	178–180	-	-

^a^ Reactions were performed with 100 mg of 50% Al_2_O_3_ as a catalyst in a stoichiometric ratio of each reaction and ethanol was compensated during the experiment for each reaction. ^b^ Isolated yields.

**Table 7 materials-16-02835-t007:** Recyclability studies of 50% Al_2_O_3_ catalyst ^a^ under solar heat.

Run	Time (min)	Yield (%)	Catalyst Amount (mg)
1	35	98	100
2	35	96	90
3	35	95	86
4	35	94	81

^a^ Reactions performed under optimized conditions in stoichiometric proportions of each reacting species with 100 mg of 50% Al_2_O_3_/PVC catalyst and a compensatory amount of ethanol in each reaction.

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
