# Peer review of "Solar-Driven Thermocatalytic Synthesis of Octahydroquinazolinone Using Novel Polyvinylchloride (PVC)-Supported Aluminum Oxide (Al2O3) Catalysts"

_materials, 2023, doi:10.3390/ma16072835_

Round 1

Reviewer 1 Report

The work “Solar-powered thermocatalytic synthesis of octahydroquinazo-2 linone using novel polyvinylchloride (PVC) supported alumi-3 num oxide (Al2O3) catalysts” by Abdulrahman I. Alharthi, Mshari Alotaibi, Amani M. Alansi, Talal F. Qahtan, Imtiaz Ali, Matar. N.Al-Shalwi and Md. Afroz Bakht submitted for consideration in MATERIALS is a trial to get octahydroquinazo-2 linone by means of thermocatalysis (PVC and alumina), solar-powered.

Well, the first point is that with the abstract it seems very complicated for how it is written, and even some odd terms like thermocatalytic, what does it mean here?

The explanation about binding energies is confusing in “binding energies that correspond to the C–C and C–H bonds in entities of –CH=CH– 248 type and the C–Cl bonds in the entities of –(CHCl–CH2)– and O=C−O bonds “. Actually, how is an O=C-O bond?

There are few typos like “octahydrquinazoline” or “Chlororo” but are minimal.

Even though Figure 7 includes a proposed mechanism please joint the arrows since it does not make sense with arrows going nowhere

In the references remove the dot after the journal titles when not necessary like in “Molecules.” And use the abbreviations such as un “Pure and Applied 424 Chemistry.” and “Angewandte Chemie International Edition.”…

Overall, if the minor points above can be addressed, the paper gives insight in the generation of quinazolinones.

Author Response

Compliance with reviewer’s and editor’s comments

Manuscript ID: materials-2216971

Journal: Materials

Title: Solar-driven thermocatalytic synthesis of octahydroquinazolinone using novel polyvinylchloride (PVC) supported aluminum oxide (Al2O3) catalysts

 Authors: Abdulrahman I. Alharthi, Mshari Alotaibi, Amani M. Alansi, Talal F. Qahtan, Imtiaz Ali, Matar. N.Al-Shalwi,  Md. Afroz Bakht

Thank you very much for the valuable suggestions and comments from the reviewers and the editor, which undoubtedly improved the quality of the revised manuscript. The manuscript has been modified as recommended by the reviewers and all changes are marked with blue fonts.  The itemized reply for the reviewer’s and editor’s comments are presented below.

Response to Reviewer #1

We thank the reviewer for his/her positive recommendation about our work and her/his constructive feedback that improved the quality of our work.

Comment -1 

  1. Well, the first point is that with the abstract it seems very complicated for how it is written, and even some odd terms like thermocatalytic, what does it mean here?

Response:

Thank you for pointing this out! The abstract has been rephrased as highlighted by the blue font on page 1.

The chemical industry is one of the main fossil fuel consumers, so its reliance on sustainable and renewable resources such as wind and solar energy should be increased to protect the environment. Accordingly, solar-driven thermocatalytic synthesis of octahydroquinazolinone using polyvinylchloride (PVC)-supported aluminum oxide (Al2O3) as a catalyst under natural sunlight is proposed in this work.

Comment -2

 2- The explanation about binding energies is confusing in “binding energies that correspond to the C–C and C–H bonds in entities of –CH=CH– 248 type and the C–Cl bonds in the entities of –(CHCl–CH2)– and O=C−O bonds “. Actually, how is an O=C-O bond?

There are a few typos like “octahydrquinazoline” or “Chlororo” but are minimal.

Response:

 Thank you for pointing this out! we modified the bonds in lines (240-241) and (246-248).

Also, the typos were modified in lines (137, 301, and 327).

 Comment -3

  1. Even though Figure 7 includes a proposed mechanism please join the arrows since it does not make sense with arrows going nowhere.

Response:

 Thank you for suggesting it!  Figure 7 on page 15 has been modified.

Comment -4

  1. In the references remove the dot after the journal titles when not necessary like in “Molecules.” And use the abbreviations such as un “Pure and Applied 424 Chemistry.” and “Angewandte Chemie International Edition.

Response:

 Thank you for pointing this out! The references were modified.

Author Response

Compliance with reviewer’s and editor’s comments

Manuscript ID: materials-2216971

Journal: Materials

Title: Solar-driven thermocatalytic synthesis of octahydroquinazolinone using novel polyvinylchloride (PVC) supported aluminum oxide (Al2O3) catalysts

 Authors: Abdulrahman I. Alharthi, Mshari Alotaibi, Amani M. Alansi, Talal F. Qahtan, Imtiaz Ali, Matar. N.Al-Shalwi,  Md. Afroz Bakht

Response to Reviewer #2

We thank the reviewer for his/her positive recommendation about our work and her/his constructive feedback that improved the quality of our work.

Comment -1 

  1. The authors declared the solar-powered thermocatalytic synthesis of octahydroquinazolinone by a novel catalyst but I didn’t find a great interest or any specific novelty in the manuscript. The authors declared the results of novel synthesis under different parameters but didn’t declare and compare the synthesis at different temperatures under different conditions. I strongly recommend doing the studies at different temperatures. Why? Because if someone wants to do the same but how much solar power (intensity/ temperature) is needed is not known. So the whole project does not lead to high scientific interest.

Response:

 Thank you for pointing this out! The reaction was performed in the month of June-July in Saudi Arabia and the amount of heat generated by the black-coated beaker is close to 80 0C in ethanol for the synthesized compounds. For the optimization, model compound (4a) also used other solvents in solar heat and also achieved their boiling temperature as measured by a thermometer. In our previous affords we compared the yield of octahydroquinzolinone derivatives at different temperatures in the conventional synthesis using a hot plate (M. A. Bakht, M. Alotaibi, A. I. Alharthi, I. Ud Din, A. Ali, A. Ali, M. J. Ahsan, Pd-HPW/SiO2    Bi-functional Catalyst: Sonochemical Synthesis, Characterization, and Effect on Octahydroquinazolinone Synthesis, Catalysts, 11(2021), 2021, 1273.  doi.10.3390/catal11111273.

Comment -2 

  1. Too much XPS detail is provided in Figure 6. The authors should summarize the whole in one Figure.

Response:

 Thank you for suggesting it! Most XPS spectra were summarized and combined in one Figure (Figure 6 on page 10). Whereas the other XPS spectra were added in the supplementary (Figure S1).

 Comment -3 

  1. NMR of the synthesized compounds should be part of supplementary data.

Response:

 Thank you for suggesting it! The NMR of the synthesized compounds is now attached in the supplementary file.

Reviewer 3 Report

In this manuscript, the authors synthesized octahydroquinazolinone derivatives through an affordable process that relies on polyvinylchloride supported aluminum oxide through sunlight exposure as a free source of heat, and 50%Al2O3/PVC catalyst performed better than any other alternative under solar heat in terms of yield and reaction time. On the whole, the experimental process is performed well, and the quality of the experimental results and discussion is adequate and reasonable. So I suggest the acceptance to this work for publication on Materials after moderate revision. However, I think the authors should consider the following questions:

 (1) In the Figure 2 and Figure 6, the reviewer did not understand how to observe the content of each component from the XRD spectrum? Why does Al2O3 contain elements C, Ca and S? Why didn't the author choose the analytically pure Al2O3 when purchasing raw material? But choose Al2O3 containing impurities?

(2) In the catalytic evaluation reaction section, “Before its use, the beaker was painted black to facilitate the absorption of the heat generated by sunlight”, the reviewer wants to know how the author ensures that the sunlight absorbed is the same every time? Is the light source used in the experiment a natural light source?

(3) Are there any other products in the products of catalytic reaction? The reviewer believes that other products should be generated during the organic reaction. In addition, how does the author calculate the yield? Please give the calculation formula.

(4) Does the state of catalyst change after catalytic reaction? For example, morphology and crystal structure, it is suggested that the author give SEM and XRD characterization.

Author Response

Compliance with reviewer’s and editor’s comments

Manuscript ID: materials-2216971

Journal: Materials

Title: Solar-driven thermocatalytic synthesis of octahydroquinazolinone using novel polyvinylchloride (PVC) supported aluminum oxide (Al2O3) catalysts

 Authors: Abdulrahman I. Alharthi1, Mshari Alotaibi1, Amani M. Alansi2, Talal F. Qahtan3, Imtiaz Ali4, Matar. N.Al-Shalwi5 and  Md. Afroz Bakht1*

Thank you very much for the valuable suggestions and comments from the reviewers and the editor, which undoubtedly improved the quality of the revised manuscript. The manuscript has been modified as recommended by the reviewers and all changes are marked with blue fonts.  The itemized reply for the reviewer’s and editor’s comments is presented below.

Response to Reviewer #3

We thank the reviewer for his/her positive recommendation about our work and her/his constructive feedback that improved the quality of our work.

Comment -1

1- In the Figure 2 and Figure 6, the reviewer did not understand how to observe the content of each component from the XRD spectrum? Why does Al2O3 contain elements C, Ca, and S? Why didn't the author choose the analytically pure Al2O3 when purchasing raw material? But choose Al2O3 containing impurities?

Response:

 Thank you for pointing this out! The presence of CaSO4 in alumina has a positive effect on the PVC. CaSO4 is an inorganic filler for PVC used to enhance its properties.

Comment -2

2- In the catalytic evaluation reaction section, “Before its use, the beaker was painted black to facilitate the absorption of the heat generated by sunlight”, the reviewer wants to know how the author ensures that the sunlight absorbed is the same every time? Is the light source used in the experiment a natural light source?

Response:

 Thank you for pointing this out!

To minimize the effect caused by the fluctuation of the sunlight irradiation, we have used more than one reactor at the same time to test each experimental parameter under the same operating condition.

Comment -3

3- Are there any other products in the products of catalytic reaction? The reviewer believes that other products should be generated during the organic reaction. In addition, how does the author calculate the yield? Please give the calculation formula.

Response: Thanks for raising important questions.

Yes, there is little extra products or reactant was available confirmed by Thin layer chromatography (TLC). The side products generated in different compounds were different and separated by repeated washing and recrystallization by ethanol and therefore depending upon the side products yield of final products affected.

Thus the product yield of final compounds was calculated as per following formula

% yield = Actual yield/ Theoretical yield X 100

Comment -4

4- Does the state of catalyst change after catalytic reaction? For example, morphology and crystal structure, it is suggested that the author give SEM and XRD characterization.

Response:

 Thank you for your suggestion!

We believe that morphology and crystal structure measurements (SEM and XRD) will not give an accurate perception of the extent of catalyst change. On the other hand, we have chosen the best catalyst in performance (50% Al2O3/PVC) and conducted a study on its recyclability. It was found that it gave an almost constant yield.

Reviewer 4 Report

In this work, the polyvinylchloride (PVC)-supported aluminum oxide (Al2O3) catalyst was prepared and used for solar-driven thermocatalytic synthesis of octahydroquinazolinone. The catalyst was well characterized; however, the manuscript still needs to further improved. I recommend a minor revision before acceptance. Here are some questions that should be addressed.

1.      For the catalyst preparation, the Al2O3 was impregnated on the surface of PVC, is there any interaction between the Al2O3 and PVC? How to avoid the aggregation of the Al2O3 particles?

2.      The names of the samples are not uniformed, such as the “50% Al2O3”, the “50% Al/PVC”, the “50%Al2O3/PVC” and the “50 wt % Al2O3/PVC”.

3.      Fig. 2, the XRD patterns shows that the prepared Al2O3/PVC catalyst exhibits “a mixture comprising 67.3% aluminum oxide, 16.7% bassanite (2CaSO4.H2O), 2.5% calcium sulfate, 10.3% aluminum oxide hydroxide (bohmite), and 3.2% aluminum hydroxide oxide hydrate (nordstrandite)”. How the contents of each samples were determined? Moreover, in Fig. 2, some of the diffraction peaks were overlapped, how to confirm the attribution of the diffraction peaks?

4.      During the reaction, what’s the role of Al2O3 and PVC? And what’s the active center for the synthesis of octahydroquinazolinone derivatives?

5.      The conclusions can be contained more characterization results.

6.      It is better to change the surface area in Table 1 to an integer, such as the “105.40” changed to “105”. Moreover, the pore radius retains one decimal place.

Author Response

Response to Reviewer #4

We thank the reviewer for his/her positive recommendation about our work and her/his constructive feedback that improved the quality of our work.

 Comment -1

1-For the catalyst preparation, the Al2O3 was impregnated on the surface of the PVC, is there any interaction between the Al2O3 and PVC? How to avoid the aggregation of the Al2O3 particles?

Response:

The BET (Shows an increase in the surface area) and XPS (Shows an increase in the adsorption sites) results confirm that there is a bonding between Al2O3 and PVC which prevents the aggregation of the Al2O3 particles.

Comment -2 

2- The names of the samples are not uniformed, such as the “50% Al2O3”, the “50% Al/PVC”, the “50%Al2O3/PVC” and the “50 wt % Al2O3/PVC”.

Response: 

Thank you for pointing this out! The names of the samples were modified.

Comment -3 

3-Fig. 2, the XRD patterns show that the prepared Al2O3/PVC catalyst exhibits “a mixture comprising 67.3% aluminum oxide, 16.7% bassanite (2CaSO4.H2O), 2.5% calcium sulfate, 10.3% aluminum oxide hydroxide (bohmite), and 3.2% aluminum hydroxide oxide hydrate (nordstrandite)”. How the contents of each sample were determined? Moreover, in Fig. 2, some of the diffraction peaks were overlapped, how to confirm the attribution of the diffraction peaks?

Response: 

Thank you for pointing this out! From the XRD measurement using software called “Match”.

Comment -4

     4-During the reaction, what’s the role of Al2O3 and PVC? And what’s the active center for the synthesis of octahydroquinazolinone derivatives?

Response: 

Thank you for pointing this out! the role of Al2O3 and PVC is explained in detail in the introduction from lines (54-74).

A catalytic active center is the Lewis acid center on the Alumina surfaces which activates the carbonyl group of the aldehyde by bonding with the oxygen of this group. Chlorine in the PVC here is an electron-withdrawing group that increases the acidity of a Lewis acid and thus promotes the activation of the carbonyl group and the initiation of the reaction. So, both the Al2O3 and PVC play an effective role to activates the carbonyl group of the aldehyde.

Comment -5

 5-The conclusions can be contained more characterization results.

Response:

Thank you for your suggestion! The conclusion was modified.

Comment -6

6- It is better to change the surface area in Table 1 to an integer, such as the “105.40” changed to “105”. Moreover, the pore radius retains one decimal place.

Response:

Thank you for your suggestion! The surface area and pore radius in Table 1 were modified.

Reviewer 5 Report

The manuscript is good and could be published as it is.

Author Response

Compliance with reviewer’s and editor’s comments

Manuscript ID: materials-2216971

Journal: Materials

Title: Solar-driven thermocatalytic synthesis of octahydroquinazolinone using novel polyvinylchloride (PVC) supported aluminum oxide (Al2O3) catalysts

Authors: Abdulrahman I. Alharthi1, Mshari Alotaibi1, Amani M. Alansi2, Talal F. Qahtan3, Imtiaz Ali4, Matar. N.Al-Shalwi5 and  Md. Afroz Bakht1*

Reviewer #5 Comments

The manuscript is good and could be published as it is.

Reviewer 6 Report

This manuscript discusses the solar-driven thermocatalytic synthesis of octahydroquinazolinone using alumina modified by polyvinylchloride. Since the main research object is the material, the results of this work correspond to the topic of the Journal «Materials». However, the manuscript needs significant revision and cannot be published without correction of the text.

Comments

1. Authors should carefully read the «Instructions for Authors» for this Journal and prepare the manuscript taking them into account.

2. Why was alumina chosen as a support, and not carbon materials, which effectively absorb sunlight? In Introduction the choice of support for polyvinyl chloride should be more carefully justified.

3. Section 2.1 should be divided into two parts: 2.1. Materials and 2.2 Catalyst characterization.

4. Section 2.1. Materials: Indicate the content of the main substance in each reagent. CAS?

5. A photo of the experimental setup should be given.

6. Figure 2: What XRD database was used to identify the phase composition of the catalysts? PDF cards?

7. What is the phase composition of alumina? It should be noted that the phase composition of alumina will also determine its surface properties. How important is this parameter in your experiments? Were such estimates made?

8. The authors claim that «and that located at 3606 cm-1 is 176 attributed to the Al−OH groups.». In this region of the IR spectrum, there are absorption bands corresponding to the stretching vibrations of the О-Н bond, even if it is water. This phrase should be written more correctly.

9. Table 1: What is the name of the unit of measure (A0)?

10. XPS data: A table with the ratios of the electronic state of the elements and their content (ratio Al:Cl) should be added to the text of the manuscript.

11. Tables 2, 6: Comments on tables should be separated from the main text.

12. Breaking tables is not allowed. Tables must be on the same page.

13. How does the catalyst change after the reaction? These data should also be given in the manuscript.

14. The main question: What is a catalytic active center? What is the role of chlorine and acid sites on the alumina surface?

In general, an interesting approach to synthesis of octahydroquinazolinone was discussed, but authors should change the text of the manuscript. I hope that my comments will be useful to the authors.

Author Response

Manuscript ID: materials-2216971

Journal: Materials

Title: Solar-driven thermocatalytic synthesis of octahydroquinazolinone using novel polyvinylchloride (PVC) supported aluminum oxide (Al2O3) catalysts

 Authors: Abdulrahman I. Alharthi1, Mshari Alotaibi1, Amani M. Alansi2, Talal F. Qahtan3, Imtiaz Ali4, Matar. N.Al-Shalwi5 and  Md. Afroz Bakht1*

Thank you very much for the valuable suggestions and comments from the reviewers and the editor, which undoubtedly improved the quality of the revised manuscript. The manuscript has been modified as recommended by the reviewers and all changes are marked with blue fonts.  The itemized reply for the reviewer’s and editor’s comments is presented below.

Response to Reviewer #6

We thank the reviewer for his/her positive recommendation about our work and her/his constructive feedback that improved the quality of our work.

Comment -2

Authors should carefully read the «Instructions for Authors» for this Journal and prepare the manuscript taking them into account.

2- Why was alumina chosen as a support, and not carbon materials, which effectively absorb sunlight? In the Introduction the choice of support for polyvinyl chloride should be more carefully justified.

Response:

Thank you for pointing this out!

The main reason for choosing alumina as catalyst support is its acidic properties which makes it an effective catalyst for the synthesis of octahydroquinazolinone derivatives which rely on the condensation of carbonyl compounds and urea in order to synthesize quinazolines. All this information was mentioned in the introduction line (53-86).  Also, its high stability compared to the carbon materials such as graphene which is susceptible to oxidative environments. Another reason is the alumina-supported catalyst has a higher pore size distribution than an alumina-supported catalyst ( https://www.mdpi.com/1996-1073/13/18/4967).

All the information on the PVC was mentioned in lines (61-73) and this study more focuses on the synthesis method (solar-driven thermo catalysis) than on catalysts.

Comment -3

3- Section 2.1 should be divided into two parts: 2.1. Materials and 2.2 Catalyst characterization.

Response: 

Thank you for your suggesting this! Section 2.1 was modified.

Comment -4

4- Section 2.1. Materials: Indicate the content of the main substance in each reagent. CAS?

Response: 

Thank you for your suggesting this! The materials content was indicated in section 2.1.

Comment -5

5- A photo of the experimental setup should be given.

Response: 

Thank you for pointing this out! A photo of the experimental setup was added as an inset for Figure 1 in the revised manuscript.

 Comment -6 

6- Figure 2: What XRD database was used to identify the phase composition of the catalysts? PDF cards?

Response: 

Thank you for pointing this out!

 Match software was used to identify the phase composition.

 Comment -7 

7- What is the phase composition of alumina? It should be noted that the phase composition of alumina will also determine its surface properties. How important is this parameter in your experiments? Were such estimates made?

Response: 

Thank you for pointing this out!

Match software was used to identify the phase composition.

However, in this study, the phase composition of alumina does not significantly affect the catalytic activity of alumina. It is clear from Table.1 and 2 that the greatest effect on the catalytic activity of alumina is the proportion of the PVC added.

Comment -8 

8- The authors claim that «…and that located at 3606 cm-1 is 176 attributed to the Al−OH groups.». In this region of the IR spectrum, there are absorption bands corresponding to the stretching vibrations of the О-Н bond, even if it is water. This phrase should be written more correctly.

Response: 

Thank you for pointing this out!

Absorption bands of 3606 cm-1 present on spectra of Al2O3, belong to oscillations of OH-groups. Similar observation mentioned in some literature [ Ref No 42]. The phrase is corrected.

 Comment -9

     9- Table 1: What is the name of the unit of measure (A0)?

Response: 

Thank you for pointing this out! The name of the unit of measure is Angstrom. The unit of measure has been modified (Å).

Comment -10

     10- XPS data: A table with the ratios of the electronic state of the elements and their content (ratio Al:Cl) should be added to the text of the manuscript.

     Response:

    Thank you for your suggestion!

 A Table with the ratios of the electronic state of the elements was added in the supplementary file (Table S1).

    Comment -11

   11- Tables 2, and 6: Comments on tables should be separated from the main text.

    Response:

    Thank you for your suggestion! Tables 2 and 6 were modified.

   Comment -12

 12- Breaking tables is not allowed. Tables must be on the same page.

  Response:

 Thank you for your suggestion! They were modified.

 Comment -13

  13- How does the catalyst change after the reaction? These data should also be given in the manuscript.

   Response:

      Thank you for pointing this out!

We have chosen the best catalyst in performance (50% Al2O3/PVC) and conducted a study on its recyclability. It was found that it gave an almost constant yield.

    Comment -14

  14- The main question: What is a catalytic active center? What is the role of chlorine and acid sites on the alumina surface?

   Response:

      Thank you for pointing this out! A catalytic active center is the Lewis acid center on the Alumina surfaces which activates the carbonyl group of the aldehyde by bonding with the oxygen of this group. Chlorine in the PVC here is an electron-withdrawing group that increases the acidity of a Lewis acid and thus promotes the activation of the carbonyl group and the initiation of the reaction. So, both the Al2O3 and PVC play an effective role to activates the carbonyl group of the aldehyde.

Round 2

Reviewer 3 Report

The authors have carefully revised the manuscript and basically made satisfactory response to me. I think the revised manuscript is suitable for publication in Materials at this state.

Author Response

Reviewers 3 round 2

Comments:

The authors have carefully revised the manuscript and basically made satisfactory response to me. I think the revised manuscript is suitable for publication in Materials at this state.

Response: Reviewer already accepted current form of MANUSCRIPT

Reviewer 6 Report

I consider that the paper has been improved according to the Reviewer's recommendations. I recommend this paper to be accepted for the publication in Journal "Materials".
But!
1. Analysis of XRD data should be done more carefully. Authors should contact highly qualified specialists in this field. Specify a PDF card for each phase from the Match database.

Sincerely, Reviewer

Author Response

Compliance with reviewer’s and editor’s comments

Manuscript ID: materials-2216971

Journal: Materials

Title: Solar-driven thermocatalytic synthesis of octahydroquinazolinone using novel polyvinylchloride (PVC) supported aluminum oxide (Al2O3) catalysts

 Authors: Abdulrahman I. Alharthi1, Mshari Alotaibi1, Amani M. Alansi2, Talal F. Qahtan3, Imtiaz Ali4, Matar. N.Al-Shalwi5 and  Md. Afroz Bakht1*

Thank you very much for the valuable suggestions and comments from the reviewers and the editor, which undoubtedly improved the quality of the revised manuscript.

Response to Reviewer #6, Round 2

Comment -1

I consider that the paper has been improved according to the Reviewer's recommendations. I recommend this paper to be accepted for the publication in Journal "Materials".
But!
1. Analysis of XRD data should be done more carefully. Authors should contact highly qualified specialists in this field. Specify a PDF card for each phase from the Match database.

Response:

 We thank the reviewer for his/her positive recommendation about our work and her/his constructive feedback that improved the quality of our work.

  1. Interesting point!

All these phases are expected as the raw material contains 10 % CaSO4 which reacts with water molecules from the aqueous medium used in sample preparation. Following reviewer suggestions, the analysis result and PDF card obtained from the match database are included below:
